# Exploration of the Safety and Solubilization, Dissolution, Analgesic Effects of Common Basic Excipients on the NSAID Drug Ketoprofen

**DOI:** 10.3390/pharmaceutics15020713

**Published:** 2023-02-20

**Authors:** Heba A. Abou-Taleb, Mai E. Shoman, Tarek Saad Makram, Jelan A. Abdel-Aleem, Hamdy Abdelkader

**Affiliations:** 1Department of Pharmaceutics and Industrial Pharmacy, Faculty of Pharmacy, Merit University (MUE), Sohag 82755, Egypt; 2Department of Medicinal Chemistry, Faculty of Pharmacy, Minia University, Minia 61519, Egypt; 3Department of Pharmaceutics and Industrial Pharmacy, Faculty of Pharmacy, October 6 University, October 6 12585, Egypt; 4Department of Industrial Pharmacy, Faculty of Pharmacy, Assiut University, Assiut 71526, Egypt; 5Department of Pharmaceutics, College of Pharmacy, King Khalid University, Abha 61441, Saudi Arabia

**Keywords:** ketoprofen, L-arginine, L-lysine, tris, basic amino acids, writhing, gastric ulcer

## Abstract

Since its introduction to the market in the 1970s, ketoprofen has been widely used due to its high efficacy in moderate pain management. However, its poor solubility and ulcer side effects have diminished its popularity. This study prepared forms of ketoprofen modified with three basic excipients: tris, L-lysine, and L-arginine, and investigated their ability to improve water solubility and reduce ulcerogenic potential. The complexation/salt formation of ketoprofen and the basic excipients was prepared using physical mixing and coprecipitation methods. The prepared mixtures were studied for solubility, docking, dissolution, differential scanning calorimetry (DSC), Fourier transform infrared spectroscopy (FTIR), in vivo evaluation for efficacy (the writhing test), and safety (ulcerogenic liability). Phase solubility diagrams were constructed, and a linear solubility (AL type) curve was obtained with tris. Docking studies suggested a possible salt formation with L-arginine using Hirshfeld surface analysis. The order of enhancement of solubility and dissolution rates was as follows: L-arginine > L-lysine > tris. In vivo analgesic evaluation indicated a significant enhancement of the onset of action of analgesic activities for the three basic excipients. However, safety and gastric protection indicated that both ketoprofen arginine and ketoprofen lysine salts were more favorable than ketoprofen tris.

## 1. Introduction

An estimated 40% of commercially available drugs and up to 90% of newly discovered drug candidates have poor water solubility [1,2]. As a result, the development of solubilization techniques, as well as the search for new hydrotropes and potential water-soluble excipients to enhance the solubility and dissolution rates of poorly soluble drugs, has been an ongoing endeavor for formulation scientists [3,4]. Ketoprofen (Figure 1) is a non-steroidal anti-inflammatory drug (NSAID) that was discovered in 1968 [5]. It is the most commonly prescribed NSAID for various acute and chronic pain conditions, such as moderate to severe dental pain and osteoarthritis [5]. Ketoprofen is sold worldwide under different brand names, including as Orudis^®^ capsules in the USA and as the over-the-counter (OTC) medication Ketofan^®^ (25 mg immediate-release tablets and 50 mg capsules) on the Egyptian market. However, poor water solubility and dissolution rates of ketoprofen have resulted in erratic drug absorption and inconsistent bioavailability, especially in the first part of the gastrointestinal tract. As a weak acid, the solubility of ketoprofen in the acidic gastric fluid is minimal [6,7].

Numerous solubilization techniques have been employed to improve solubility and dissolution rates of different water-insoluble drugs. These techniques include particle size reduction, solid dispersion, complexation, salt formation, cocrystallization, and nanoparticle encapsulation [8,9,10]. In addition, many water-soluble excipients have been used to improve the solubility and bioavailability of poorly soluble drugs. These include water-soluble macromolecules and hydrophilic polymers, such as polysaccharides, polyvinylpyrrolidone, polyethylene glycol, and cyclodextrins [10]. While these excipients have successfully enhanced the solubility of many drugs, their solubilizing capacity can be limited and require that they be used in large amounts, which can raise toxicological and regulatory concerns [3,10]. Low-molecular-weight excipients, such as urea and sugars, have also been extensively investigated. However, their solubilizing capacity is limited due to both their chemical neutrality and their lack of sufficient binding sites and ionizable groups [11].

In recent years, there has been a growing interest in investigating and utilizing amino acids due to their safety and tolerability. Amino acids are classified as GRAS (Generally Recognized as Safe) and are used as dietary supplements [4]. In addition, amino acids have been successfully used to solubilize both ionizable and non-ionizable drugs. They are small molecules with diverse chemical structures, and can be broadly classified into mainly amphoteric (e.g., glycine and alanine), acidic (e.g., aspartic acid and glutamic acid), or basic (e.g., arginine and lysine) amino acids (Figure 1). Additional side chains, such as hydroxyl and sulfhydryl groups, can boost their solubilizing capacity [4,12].

Ketoprofen-L-lysine can exist in salt or cocrystal forms, depending on the preparation method. Both forms have enhanced dissolution characteristics, but the bitterness scores for these two forms of ketoprofen-L-lysine were higher than that of the parent drug [13]. In another study, ketoprofen–tromethamine was prepared by a coprecipitation method, resulting in a new crystalline state with significantly enhanced solubility and dissolution rates [14].

Tromethamine (also known as tris aminomethane) is a basic excipient and a widely used buffering agent in biochemistry and protein assays. Tris has been used to form water-soluble salts from weak acids such as ketorolac and nimesulide [15].

This study explored the impact of three basic excipients (lysine, arginine, and tromethamine, or tris) with different basicity and pKa values (Figure 1) on the solubility, dissolution rates, and analgesic efficacy of ketoprofen, as well as ulcer side effects. The aim was to rank and showcase any particular advantages of these basic excipients in improving the biopharmaceutical properties and safety profile of the NSAID drug ketoprofen.

The specific objectives of the study included the formation of solid dispersions and physical mixtures, the construction of phase solubility diagrams, thermal and dissolution studies, spectral and docking analysis, analgesic evaluation using the writhing test in mice, gastric ulcer liability, and histopathological examination.

## 2. Materials and Methods

### 2.1. Materials

Ketoprofen was provided by Pharco Pharmaceuticals (Alexandria, Egypt). L-arginine was purchased from Fluka AG (Buchs, Switzerland). L-lysine, tris, and sodium lauryl sulfate were obtained from Sigma-Aldrich (London, UK), and Ketofan^®^ capsules were supplied by Amrya Pharmaceuticals (Amrya, Alexandria, Egypt). Empty hard gelatin capsules of size 0 were purchased from Isolab Laborgeräte (GmbH, Am Dillhof, Germany).

### 2.2. Preparation of Ketoprofen-Excipients Physical and Dispersed Mixtures

#### 2.2.1. Physical Mixtures

Ketoprofen-L-lysine, L-arginine, and tris physical mixtures (PM) were prepared separately by weighing an equivalent molar weight in milligrams. The drug-excipient mixture was then thoroughly mixed in a porcelain dish for 2–3 min using a spatula and sieved through a 125-µm sieve.

#### 2.2.2. Coprecipitated Mixtures of Ketoprofen:L-lysine, Ketoprofen:L-arginine, and Ketoprofen:tris

To prepare coprecipitated mixtures of ketoprofen with L-lysine, L-arginine, and tris, specific weights (in mg) equivalent to the molecular weight of ketoprofen were dissolved in 20 mL of methanol. Accurate weights (in mg) equivalent to the molecular weights of the basic amino acids (L-lysine and L-arginine) and tris were dissolved individually in 10 mL of distilled water. The methanolic solution of ketoprofen and the aqueous solutions of the basic excipients were mixed in a porcelain dish with a 100-mL capacity. The porcelain dish was placed on a hot plate stirrer (LabTech, Daihan, Korea), adjusted to 80 °C, and left until complete evaporation. The resulting powder was ground in a mortar and pestle and passed through a 125 µm sieve.

### 2.3. Equilibrium Solubility Studies

Excess amounts of ketoprofen were added to various solutions containing different concentrations of the basic excipients (0, 0.1, 0.2, 0.4, 0.5, 1, 2, and 3% *w/v*) of L-arginine, L-lysine, and tris. These mixtures were placed in a thermostatic shaking water bath (Shel Lab water bath, Sheldon Cornelius, OR, USA) at 37 °C ± 0.5 °C, rotating at a speed of 120 strokes per minute. The samples were left for 48 h to attain equilibrium; aliquots (4 mL) were withdrawn, filtered, and measured spectrophotometrically at λ_max_ = 260 nm using a UV-visible spectrophotometer (JENWAY-Model 6305, Chelmsford, UK). The solubility data (µg/mL) were obtained from the standard calibration curve with acceptable linearity (R^2^ = 0.9955). The solubility constant (K) was calculated from the slope of the phase solubility diagram obtained from the regression line of solubility (µg/mL) versus concentration (mM) plots [9,15], as shown in the following equation:(1)$$K=\frac{Slope}{Intercept*\left(1-slope\right)}$$

### 2.4. Differential Scanning Calorimetry (DSC) and Fourier Transfer Infrared Spectroscopy (FTIR)

Samples of ketoprofen, arginine, lysine, tris, physical mixtures (PM), and coprecipitated mixtures were weighed (2–4 mg) and placed in aluminum pans. The DSC Mettler Toledo Star System (Mettler Toledo, Zürich, Switzerland) was used to gradually increase the temperature from 30 to 300 °C at a rate of 10 °C/min, calibrated with an indium standard, and using nitrogen as a purging gas. A Thermo Scientific Nicole IS 10 FTIR spectrophotometer (Waltham, MA, USA) was used to compress potassium bromide samples into discs using a 10-ton hydraulic press. The samples were scanned 16 times from 400 to 4000 cm^−1^, and data were collected using Omnic software from Thermo Scientific in Waltham, MA, USA.

### 2.5. In Vitro Dissolution

In vitro dissolution studies were conducted on two dissolution media. The first dissolution medium consisted of simulated gastric fluid (pH 1.2, 900 mL) containing 1% *w*/*w* sodium lauryl sulfate (SLS) for the first two hours. Then, in the same flask, the pH of the medium was increased to 6.8 using dibasic sodium phosphate for an additional three hours to simulate intestinal fluid. The dissolution media were agitated using USP apparatus 2 at 50 rpm and a temperature of 37 °C. Ketoprofen powder, PM, and Coppt dispersed mixtures weighing 20 mg (or equivalent to 20 mg of ketoprofen) were filled into hard gelatin capsules of size 0 (Isolab Laborgeräte, GmbH, Am Dillhof, Germany), placed in dissolution sinkers, and transferred to dissolution flasks. A 5 mL sample was withdrawn at specified intervals and replaced with another 5 mL of fresh dissolution medium. The samples were analyzed spectrophotometrically, as previously described in Section 2.3.

### 2.6. Molecular Docking

Molecular docking studies were performed with the Molecular Operating Environment (MOE) 2014.09 software (Chemical Computing Group, Montreal, QC, Canada) to predict the stability and possible orientation of various bases on the surface of ketoprofen. The 3D structure of ketoprofen was constructed using the builder interface, and its energy was minimized to an RMSD (root mean square deviation) gradient of 0.01 kcal/mol using the QuickPrep tool in the MOE software. Similarly, the 3D structures of arginine, lysine, and tromethamine were built using the MOE builder, and their energies were minimized. The three bases were docked onto the surface of ketoprofen using an induced-fit docking protocol with the Tri-angle Matcher method and dG scoring system for pose ranking. After a visual assessment of the resultant docking poses, those with the highest stability and lowest binding free energy values were selected and reported.

### 2.7. In Vivo Studies

#### 2.7.1. Writhing Assay

Mice weighing between 25 and 30 g were used in the experiment. The ability of ketoprofen and the prepared coprecipitated mixtures of ketoprofen with the three basic excipients (tris, L-lysine, and L-arginine) to inhibit acetic acid-induced writhing was assessed as previously described [11]. The mice were divided into five groups, as outlined in Table 1. A dose of 50 mg/kg or its equivalent was dispersed in an aqueous solution containing 0.25% carboxymethyl cellulose (CMC) to make the tested solutions (2 mg/mL). An accurate sample (0.5 mL) of the tested solutions was administered orally through a gastric tube. After the dose was administered, 30 µL of diluted acetic acid solution (0.6% *v*/*v*) was injected intraperitoneally into the animals. Induced writhes were counted for 20 min.

#### 2.7.2. Indomethacin-Induced Ulcer

Male albino rats were fasted for 24 h and given access to water. They were divided into six groups of five rats each. The positive control group received a single oral dose of indomethacin (30 mg/kg) through a gastric tube, while the control group received saline. The remaining four groups were given a single oral dose of 50 mg/kg of ketoprofen or its equivalent in K:tris, K:lysine, and K:arginine Coppt mixtures. Four hours after dosing, the animals were sacrificed, and their stomachs were dissected, flushed with saline, and opened for inspection of ulcer formation [16].

The ulcers were counted and quantified by pinning the stomach on a piece of flat cork and scoring the ulcers using a dissecting microscope. The area of mucosal damage (ulcer) was expressed as a percentage of the total surface area of the mucosal surface of the stomach [16].

### 2.8. Histopathological Documentation

The dissected stomachs from the control and treated groups were fixed in 10% formalin-buffered saline for several days, dehydrated, embedded in paraffin blocks, and then sectioned into 5 μm-thick slices. As previously reported, the final sections were stained with H&E stain for microscopic examination and imaging [17,18]. The aggregation of polysaccharides was visualized using periodic acid Schiff (PAS) staining [19].

## 3. Results and Discussion

### 3.1. Solubility Studies

The results of the equilibrium solubility of ketoprofen in the presence of increasing concentrations (*w/v* %) of the three basic excipients (L-lysine, L-arginine, and tris) are shown in Figure 2A. Phase solubility curves of ketoprofen with the three basic excipients were constructed (Figure 2B–D) to determine solubility type. The three basic excipients, tris, L-lysine, and L-arginine, at a concentration of 3% *w/v*, significantly (*p* < 0.05) improved the solubility of ketoprofen by 4, 4.65, and 6.8-fold, respectively. Arginine showed superior solubilization capacity compared to the other two basic excipients (Figure 2A). This solubility enhancement can be attributed to the electrostatic interaction and alkalinizing effects of the stagnant diffusion layer around dissolved particles of the basic excipients in the solvent, which increased the ionization of the weak acid drug [20]. Arginine (pKa = 12.48) is the strongest basic excipient compared to tris (pKa = 8) and lysine (pKa = 10). Hence, the diffusion layer becomes more alkaline and more ionization occurs, favoring the solubilization of the weak acid ketoprofen. Figure 2B-D shows the phase solubility curves (solubility (mM) against concentrations (mM) of the basic excipients). Tris obtained solely a linear relationship (AL). In contrast, nonlinear relationships were observed with arginine and lysine.

Similarly, the solubility of ketoprofen in the prepared physical mixtures (PM) and coprecipitated ketoprofen with tris, L-lysine, and L-arginine was enhanced (Figure 3). Ketoprofen’s solubility increased significantly (*p* < 0.05), and this increase depended on the type of excipient and the preparation method of the solid dispersion. For example, coprecipitated dispersed mixtures demonstrate superior enhancement in solubility compared to physical mixtures. Coprecipitated mixtures generate drug particles with less particle size due to the solvent effect, in addition to generating more intimate contact and interactions with the basic excipients; hence, higher solubility can be achieved [1].

### 3.2. FTIR and DSC Studies

FTIR spectroscopy and DSC thermal analysis were used to detect possible physicochemical interactions between ketoprofen and the three basic excipients under investigation. Figure 4A shows the FTIR spectra of ketoprofen, the three basic excipients, and their physical and coprecipitated mixtures. Specific IR absorption bands of pure ketoprofen detected at 1610 cm^−1^ and 1684 cm^−1^ were due to stretching of the ketone group and the carboxylic carbonyl group (C=O), respectively [14]. The characteristic peaks at their assigned wavenumbers were simply additive to the FTIR spectra of the three basic excipients, indicating that no observable physicochemical interactions could be identified with the physical mixtures. In contrast, the vibrational bands of the keto and carbonyl groups of ketoprofen were broadened and shifted for ketoprofen and tris, ketoprofen and L-lysine, and ketoprofen and L-arginine, suggesting hydrogen bonding formation and electrostatic interactions with the basic/cationic excipients [21]. Similarly, DSC analysis revealed the complete disappearance of ketoprofen melting from both physical and coprecipitated ketoprofen mixtures with L-lysine and tris. This indicated the presence of both physicochemical and electrostatic attraction. In contrast, a weak melting transition was found in K:L-arginine PM, but a complete disappearance of K melting was observed in the coprecipitated mixtures.

### 3.3. Dissolution Studies

For drugs with poor solubility, determined dissolution rates are both a regulatory requirement and essential for distinguishing newly developed formulations. The dissolution medium should mimic physiological fluids and conditions [22]. To determine the exact amount of ketoprofen used for the in vitro dissolution study under sink conditions, equilibrium solubility was measured in different simulated gastric fluids (0.1 M HCl) containing three different concentrations (0.1%, 0.5%, and 1% *w/v*) of sodium lauryl sulfate as a surfactant. Sodium lauryl sulfate is an anionic surfactant that was selected because it mimics the anionic natural surfactants/bile salts in gastric fluid. To both prevent the surface flotation of drug particles and simulate in vivo performance, it is crucial to wet the dispersed particles prior to dissolution. The surface tension of gastric fluid is considerably lower than that of water, suggesting the presence of surfactants in this region [23]. Ketoprofen solubility was 75, 127.5, 150, and 190 µg/mL for HCl solutions containing SLS of 0.1%, 0.5%, and 1% *w/v*, respectively. Therefore, an acid dissolution medium with 1% SLS was selected to ensure sink conditions.

This study adopted both acidic gastric conditions and a physiological pH of 6.8 to simulate intestinal pH. The first two hours of dissolution were studied at an acidic pH because the solubility of ketoprofen (a weakly acidic drug with a pKa of 4.4) was very low (0.1 mg/mL), the pH was significantly lower than the pKa of ketoprofen, and the drug was available in a unionized form. In contrast, the solubility of the drug at pH 6.8 (where the drug predominantly exists in ionized forms) was evaluated to determine the capacity of the three basic excipients to improve the dissolution rate under gastric conditions.

Figure 5 shows the dissolution profiles of ketoprofen from the prepared physical and dispersed mixtures, and Table 2 presents three dissolution parameters: the time required for the dissolution of 50% of ketoprofen (T50%) and relative dissolution rates at 60 min and 300 min (RDR_60_ and RDR_300_, respectively). Slow and incomplete dissolution was recorded for ketoprofen powder over 300 min, with only 50% of the drug dissolving in 240 min. In contrast, Ketofan^®^ capsules showed nearly doubled RDR_60_ and RDR_300_ dissolution parameters. Similarly, physical mixtures of ketoprofen with the three basic excipients enhanced the dissolution parameters by 1.26 to 1.74-fold and shortened the T50% value to 120 and 180 min, respectively, compared to the T50% of 240 min recorded for ketoprofen powder.

Compared to physical mixtures, superior dissolution rates were recorded for coprecipitated mixtures. For example, K:tris, K:lysine, and K:arginine coprecipitates recorded T50% of 120, 60, and 30 min, respectively, compared to 180, 120, and 120 min estimated for K:tris, K:lysine, and K:arginine physical mixtures, respectively. These results indicated that the preparation technique of the dispersed mixtures made a marked difference in dissolution rates.

Furthermore, L-arginine and L-lysine appear superior to tris in terms of their capacity to improve the in vitro dissolution rates of ketoprofen. L-arginine (pKa = 12.48) is the strongest base compared to the other two basic excipients, L-lysine (pKa = 10.79) and tris (pKa = 7.8). The stronger the base, the faster the in vitro dissolution rate can be recorded. This is due to the faster alkalinization of the diffusion layer surrounding the drug particles, as well as the increasing ionization of the acidic drug in this diffusion layer [20]. Additionally, these results correlate well with the solubility studies that demonstrated the following order: L-arginine > L-lysine > tris.

### 3.4. Molecular Docking

Several methods could be utilized to establish the ability of the three basic excipients to form a salt with ketoprofen. Fundamentally, the difference in acid dissociation constants (pKa (base)–pKa (acid)) for ketoprofen and the three basic excipients is widely used to predict the possibility of cosolvation experiments producing a cocrystal or a salt. A pKa difference greater than 3 suggests a salt formation, while values less than 0 suggest that cocrystal is the predominant form [24,25,26,27]. An acidic pKa of 4.39–4.45 [28] for the propionic acid proton of ketoprofen gives a difference of 8, 6.4, and 3.4 with arginine, lysine, and tromethamine, respectively (Figure 1). This supports the previous findings of salt formation between ketoprofen and the three bases.

Additionally, Hirshfeld surface analysis, a tool for visualizing crystal structure interactions (Spackman & Jayatilaka, 2009), of ketoprofen crystals revealed that carboxylic oxygens are the most likely sites for interactions in ketoprofen (shown red areas in Figure 6A). The same conclusion was reached with theoretical docking of the three bases on the surface of ketoprofen using MOE software, suggesting the construction of small stable complexes, as shown in Figure 6B. In the case of arginine, the complexes created showed the proximity of the basic guanidine NH_2_ with the highest pKa to the carboxylic group. The stability of such complexes, and their observed proximity, may favor the potential proton transfer between the ketoprofen acidic group and the guanidine amino group of arginine in a salt formation process. Recently, the salt formation between ketoprofen and tromethamine was confirmed [14]. A salt formation between ketoprofen and lysine was also described, substantiating our assumptions [13].

### 3.5. In Vivo Studies

#### 3.5.1. Writhing Assay

A writhing assay was employed to assess the onset of analgesic activities of the drug alone and the coprecipitated drug mixtures with three basic excipients (tris, L-lysine, and L-arginine) within 20 min. In the current study, both the number of writhes and percentage (%) of writhing inhibition were recorded for the untreated, ketoprofen-, K:tris Coppt-, K:L-lysine Coppt-, and K:L-arginine-treated groups, as illustrated in Figure 7A,B. The number of writhes for the ketoprofen-treated group decreased from 46 to 32, with 30% inhibition. In contrast, the numbers of writhes recorded for the K:tris, K:L-lysine, and K:L-arginine coprecipitated mixture groups were 3, 8, and 10, respectively, with percentage inhibitions of 91%, 82%, and 78%, respectively. These findings suggest that these basic excipients have promising potential to quickly enhance analgesic activity when compared to the drug alone. This is due to their improved solubility and in vitro dissolution rates. Notably, it is worth mentioning that this in vivo study did not significantly correspond with previously mentioned in vitro dissolution studies, where L-arginine showed superior potential for enhancing both solubility and dissolution rates.

K:tris Coppt demonstrated statistically significant inhibition in the number of writhes compared to both K:L-lysine and K:L-arginine Coppt, while the latter two showed significant reductions in the number of writhes (8 and 10, respectively) and percentage inhibition (82% and 79%, respectively). However, no statistically significant differences (*p* > 0.05) were identified for either L-arginine or L-lysine in reducing the number of writhes. Similar results were reported elsewhere for the NSAID drug nimesulide [15]. Nimesulide tris Coppt outperformed the amorphous mixture of nimesulide and PVP K30 in terms of analgesic activity and time to onset of action [15].

In another study, the ketoprofen lysine salt demonstrated a more rapid and complete absorption than the acid form of ketoprofen. Peak plasma concentration for the ketoprofen lysine salt was attained in 15 min, compared to 60 min for the acid form [29]. Additionally, it was reported that the ketoprofen lysine salt demonstrated analgesic activity two times stronger than ketoprofen, as well as a higher LD_50_ [30]. 

The writhing assay was also used to assess the onset of analgesic activity of nimesulide, a poorly soluble drug. Nimesulide alone inhibited writhing by approximately 22%. In comparison, the more water-soluble form of the drug prepared in an inclusion complex with β-cyclodextrin in a ratio of 1:4 showed a percentage inhibition of 54.5% at 20 min [31]. The nimesulide-tris complex showed a superior reduction in the number of writhes compared to the nimesulide-polyvinylpyrrolidone (PVP) K30 and nimesulide-polyethylene glycol 4000 complexes [15]. Several reports have indicated that, in addition to improving the solubility of poorly soluble drugs, tris can act as a permeability enhancer and alter membrane permeability [32,33,34].

#### 3.5.2. Indomethacin-Induced Ulcer

NSAIDs cause gastric toxicity, including gastric ulcers. Indomethacin, a commonly used NSAID, is often used as a model drug for inducing gastric ulcers in rats due to its high ulcerogenic index [16,35]. Indomethacin is a potent inhibitor of prostaglandin and can cause significant damage to the gastric mucosa [36]. This study aimed to determine if coprecipitated mixtures of ketoprofen and three basic excipients, which improved solubility and bioavailability, could reduce the gastrointestinal side effects of ketoprofen.

Figure 8 shows stomachs pinned on corkboards to emphasize the location and number of ulcers in the negative and positive (indomethacin) groups, as well as the groups treated with ketoprofen and coprecipitated mixtures of ketoprofen and basic excipients. The indomethacin-treated group (the positive control) had the highest number of ulcers, with nine ulcers recorded. The number of ulcers in the ketoprofen group was reduced to about one-third of that of the indomethacin group, as indomethacin is more potent at causing gastric ulcers [36]. There were no statistically significant (*p* > 0.05) differences in the number of ulcers between the K:tris and ketoprofen groups.

Interestingly, the K:lysine and K:arginine coprecipitated mixtures produced significantly fewer ulcers than ketoprofen alone. These results are consistent with recent reports [5]. Ketoprofen lysine salt has been shown to reduce ulcer side effects compared to the acidic form of ketoprofen [37]. This is likely due to the residual amino groups of L-lysine and L-arginine, which act as carbonyl scavengers; they also offer protection against oxidative damage to the gastric mucosa by providing indirect antioxidant effects and increasing the levels of glutathione S-transferase P at the cellular level in the gastric mucosa [5]. Additionally, L-lysine and L-arginine have been reported to both enhance mucosal integrity and have gastroprotective effects through nitric oxide (NO) donation [37,38].

#### 3.5.3. Histopathological Studies

Figure 9a–e and Figure 10a–e display histopathological documentation of gastric tissues for the control, ketoprofen, ketoprofen:tris coprecipitate, ketoprofen:lysine coprecipitate, and ketoprofen:arginine coprecipitate at low magnification (x100) and high magnification (x400) lenses. The normal control group exhibited intact mucosa (double-headed arrow), healthy surface epithelium (thin black arrow), intact normal gastric glands (white arrows), and normal submucosa (Figure 9a). At higher magnification, the normal control group showed healthy surface epithelium with normal integrity (thin black arrow) and intact normal gastric glands (white arrows) (Figure 10a). In contrast, the ketoprofen-treated group exhibited gastric mucosa (double-headed white arrow) with sporadic superficial degermation and desquamation of the surface epithelium (red arrows). Additionally, degenerative changes and shrinkage of the gastric glands (thick black arrows) were observed (Figure 9b). Figure 10b shows superficial degermation and desquamation of the surface epithelium (red dotted arrows). Furthermore, degenerative changes and shrinkage of gastric glands (thick black arrows) were recorded for the ketoprofen-treated group (Figure 10b).

Figure 9c-1,-2 shows gastric mucosa (double-headed white arrow) with ulcerated regions (red arrows) and slight degeneration of glands (thick black arrow) in the K:tris-treated group. Ulcerated surface epithelium (red dotted arrows) with slight degeneration and atrophy of gastric glands (thick black arrow) was recorded for the same group and detected at higher magnification in Figure 10c-1,-2.

Figure 9d shows normal, intact mucosa (double-headed arrow), maintained surface epithelial integrity (black arrows), and gastric glands (white arrows) for the K:lysine group. Maintained surface epithelial integrity (black arrows) and intact gastric glands (white arrows) were recorded at higher magnification (Figure 10d).

Figure 9e shows intact, healthy mucosa and possible protection against ketoprofen-induced superficial ulceration (black arrows) for the K:arginine group. In Figure 10e, intact, healthy mucosal surfaces (black arrows) and normal gastric glands (white arrows) were recorded at higher magnification.

These findings correlate significantly with the ulcer indices shown in Table 3, and, in addition to their enhanced solubilization for ketoprofen, confirm the gastroprotective effect and safety benefits of the two basic amino acids L-lysine and L-arginine.

## 4. Conclusions

This study highlighted the role of three basic excipients (tris, L-lysine, and L-arginine) as potential solubilizers, as well as their capacity to form salts with the non-steroidal anti-inflammatory drug ketoprofen. The three basic excipients were superior in potentiating and advancing analgesic activities due to both their penetration-enhancing activities and their enhanced solubility and dissolution rates of the weak acid drug. However, only L-arginine and L-lysine demonstrated gastric protection against ketoprofen-induced ulcers and erosion of the gastric mucosa. This study recommends L-arginine and L-lysine as promising agents for promoting the analgesic and safety profiles of classical NSAIDs.

## Figures and Tables

**Figure 1 pharmaceutics-15-00713-f001:**
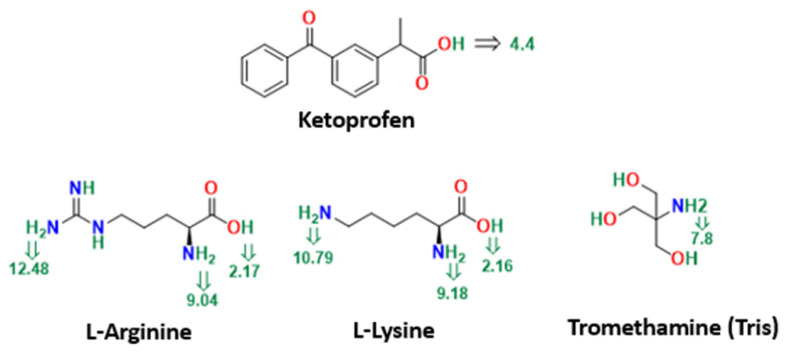
Structures and pKa values of ketoprofen, tris, L-lysine, and L-arginine.

**Figure 2 pharmaceutics-15-00713-f002:**
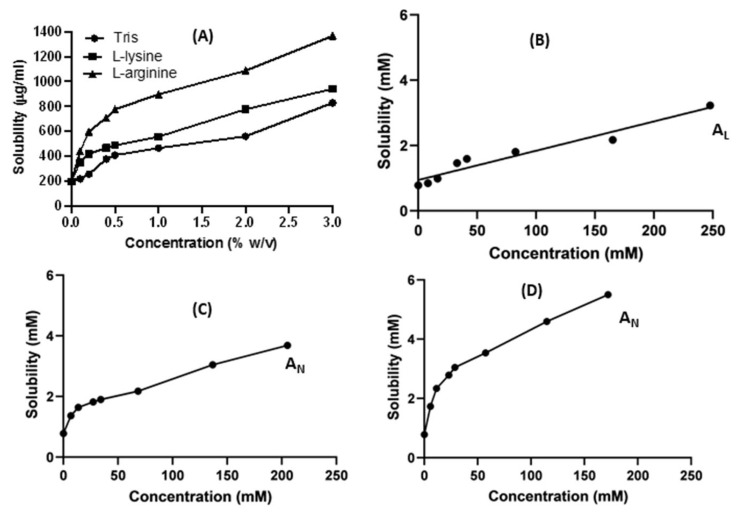
(**A**) Solubility curves µg/mL versus concentration of the three basic excipients (% *w/v*). Phase solubility curves (mM) versus concentrations of the three basic excipients (mM): (**B**) tris, (**C**) L-lysine, and (**D**) L-arginine.

**Figure 3 pharmaceutics-15-00713-f003:**
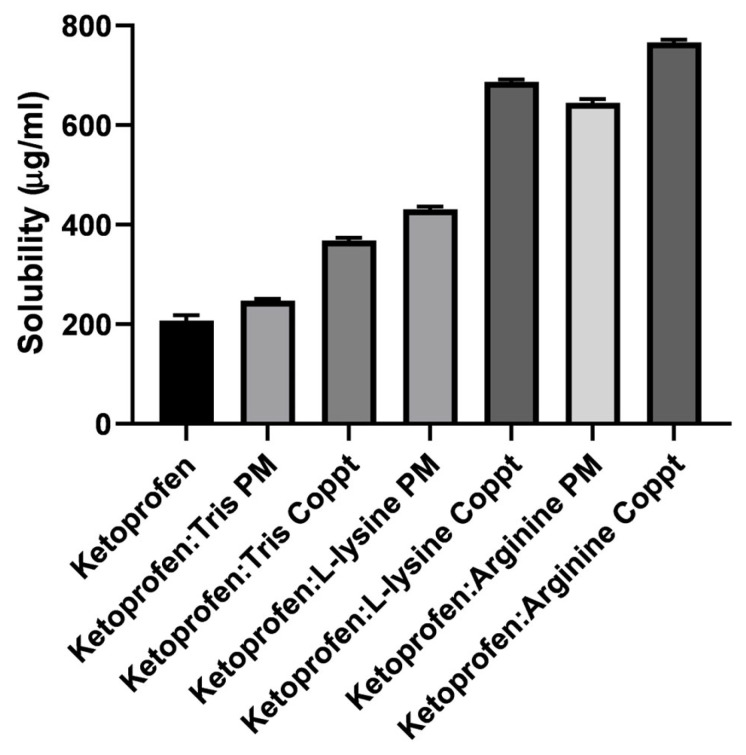
The solubility of ketoprofen in the prepared physical and coprecipitated mixtures.

**Figure 4 pharmaceutics-15-00713-f004:**
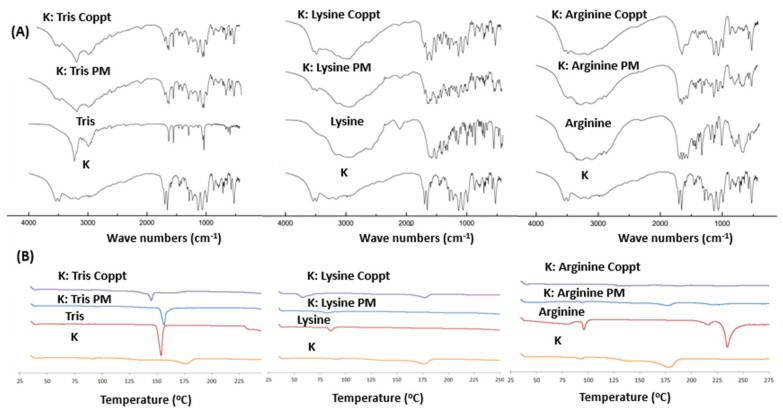
(**A**) FTIR spectra and (**B**) DSC thermograms of ketoprofen (K), basic excipients (tris, lysine, and arginine), physical mixtures (PM), and coprecipitated dispersed mixtures (Coppt).

**Figure 5 pharmaceutics-15-00713-f005:**
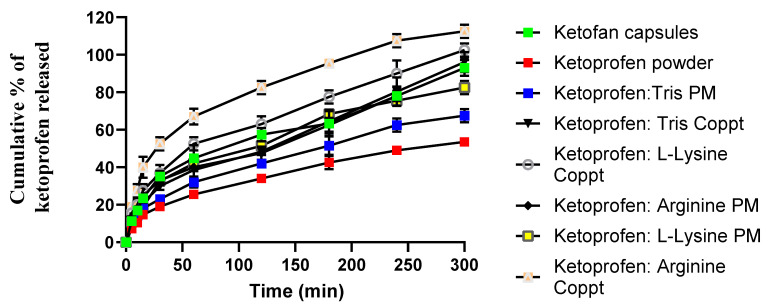
Profiles of in vitro dissolution of ketoprofen powder, commercial capsules, and physical and dispersed mixtures with lysine, arginine, and tris. For the first two hours, simulated gastric fluid (pH = 1.2) was employed, and for the remaining three hours, simulated intestinal fluid (pH = 6.8) was used.

**Figure 6 pharmaceutics-15-00713-f006:**
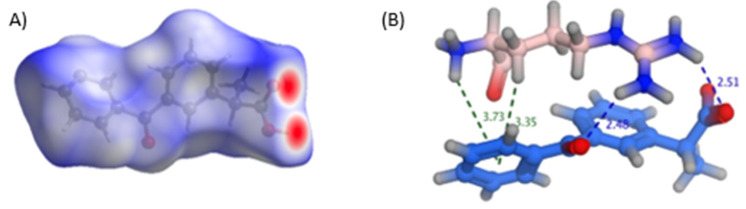
(**A**) Hirshfeld surface analysis of ketoprofen crystals and (**B**) possible interactions formed upon docking arginine onto the ketoprofen surface.

**Figure 7 pharmaceutics-15-00713-f007:**
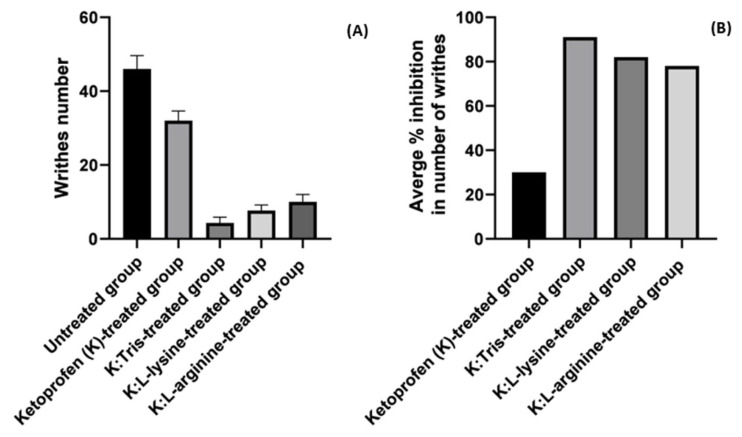
(**A**) The number of writhes and (**B**) average percentage (%) inhibition for ketoprofen and ketoprofen with respect to the three basic excipient coprecipitated mixtures.

**Figure 8 pharmaceutics-15-00713-f008:**
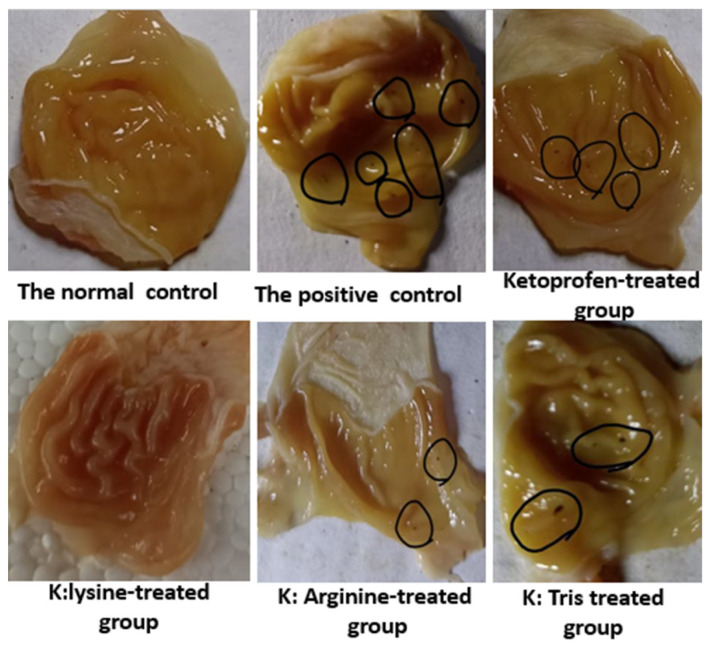
Pinned stomachs on corkboards highlighting the position and number of ulcers (encircled in black lines) for the negative and positive control (indomethacin), ketoprofen (K), K:lysine, K:arginine:K:tris coprecipitated mixtures.

**Figure 9 pharmaceutics-15-00713-f009:**
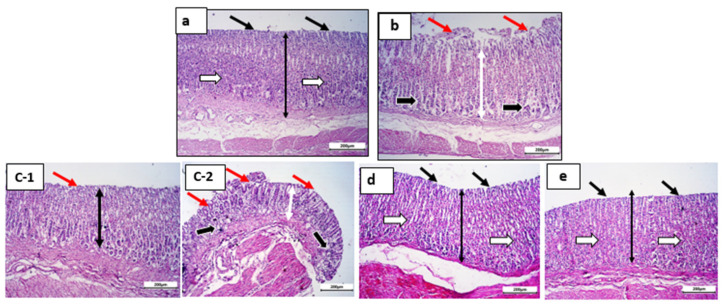
Histological sections from rat stomachs: (**a**) control group, (**b**) ketoprofen-treated group, (**c**-**1**, **c**-**2**) ketoprofen:tris Coppt, (**d**) ketoprofen:L-lysine Coppt, and (**e**) ketoprofen:arginine Coppt stained by H&E and photographed at low power x 100 (bar = 200 µm) and x 400 (bar = 50 µm).

**Figure 10 pharmaceutics-15-00713-f010:**
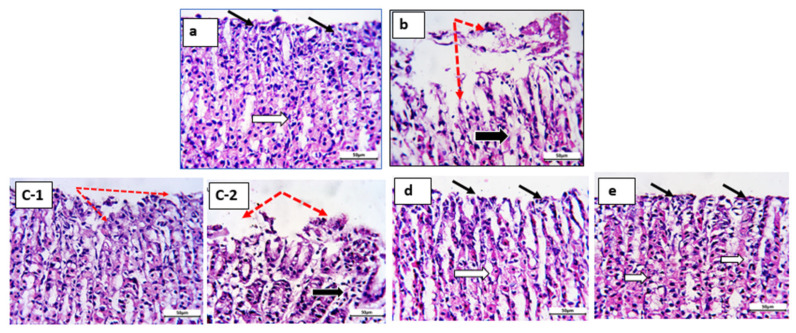
Histological sections from rat stomachs: (**a**) control, (**b**) ketoprofen-treated group, (**c**-**1**, **c**-**2**) ketoprofen:tris Coppt, (**d**) ketoprofen:L-lysine Coppt, and (**e**) ketoprofen:arginine Coppt stained by H&E and photographed at high power x 400 (bar = 50 µm).

**Table 1 pharmaceutics-15-00713-t001:** Different groups and treatments received in the writhing assay.

Group	Treatment
I	Solution of 0.25% CMC (Untreated)
II	Ketoprofen (K) suspended in 0.25% CMC
III	K:tris dispersed in 0.25% CMC
IV	K:L-lysine dispersed in 0.25% CMC
V	K:L-arginine dispersed in 0.25% CMC

**Table 2 pharmaceutics-15-00713-t002:** Dissolution parameters were measured for ketoprofen, commercial capsules, and ketoprofen with respect to excipient mixtures. * T50% denotes the time required for 50% of the initial amount to be dissolved; ** and *** denote relative dissolution rates of 60 and 300 min, respectively.

Formulation	T50% (min) *	RDR_60_ **	RDR_300_ ***
Ketoprofen powder	240	-	-
Ketofan capsule	120	2	1.75
K:tris PM	180	1.28	1.26
K:tris Coppt	120	1.6	1.8
K:lysine PM	120	1.6	1.5
K:lysine Coppt	60	2.08	1.92
K:arginine PM	120	1.74	1.74
K:arginine Coppt	30	2.68	2.07

**Table 3 pharmaceutics-15-00713-t003:** Ulcerogenic potential (number of ulcers) and ulcer indices for the positive control (indomethacin), ketoprofen (K), K:lysine, K:arginine:K:tris coprecipitated mixtures.

**Test Substance**	**Ulcer Number**	**Ulcer Index**
Control	0 ± 0.0	0
Indomethacin	8.66 ± 0.88 ^a^	7.8 ^a^
Ketoprofen	3.33 ± 0.66 ^a,b^	3.59 ^a,b^
Ketoprofen:tris	2.00 ± 0.0 ^b^	1.1 ^a,b,c^
Ketoprofen:lysine	1.0 ± 0.33 ^b,c^	0.55 ^a,b,c^
Ketoprofen:arginine	0.53.00 ± 0.17 ^b^	0.33 ^a,b,c^

The data are presented as the mean ± SD of six animals. A one-way ANOVA test followed by a Tukey–Kramer post hoc test was used for multiple comparisons. ^a^ Denotes a significant difference from the control group (*p* < 0.05). ^b^ Represents a significant difference from the indomethacin group (*p* < 0.05). ^c^ Indicates a significant difference from the ketoprofen group (*p* < 0.05).

## Data Availability

Upon request.

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
