# Peer review of "Exploration of the Safety and Solubilization, Dissolution, Analgesic Effects of Common Basic Excipients on the NSAID Drug Ketoprofen"

_pharmaceutics, 2023, doi:10.3390/pharmaceutics15020713_

Round 1

Reviewer 1 Report

The paperExploration of the solubilizing, dissolution, analgesic effects and safety of common basic excipients on the NSAID drug ketoprofen” by  Heba A. Abou-Taleb, Mai E Shoman, Tarek Saad Makram, Jelan A. Abdel-Aleem and Hamdy Abdelkader reports  the complexation/salt formation of ketoprofen with the basic excipients (Tris, L-lysine and L-arginine). The mixtures were prepared using physical mixing and coprecipitation methods. The prepared mixtures were studied for solubility, docking, dissolution, DSC, FTIR, in vivo evaluation for efficacy (writhing test) and safety (ulcerogenic liability).

The following observations should be applied:

1.      Abstract. Line 23: “FIR” must be replaced with FTIR.

2.      Material and methods: Physical mixtures: Lines 90-91: “Drug-excipient mixed thoroughly in a porcelain dish for 2-3 minutes using a spatula” is not enough to obtain a homogeneous mixture and especially a complex/salt formation. Please explain why used this method.

3.      Material and methods: Coprecipitated dispersed mixtures: Lines 93-94: please specify the composition of the mixtures.

4.      Material and methods: Equilibrium solubility studies: Lines 100-101: “Excess amounts of ketoprofen were added to different solutions containing increasing concentrations (0, 0.1, 0.2, 0.4, 0.5, 1, 2 and 3% w/v)”. Increasing concentrations of what? Please specify.

5.      Material and methods: Equilibrium solubility studies: The whole paragraph needs to be rewritten. It must contain in detail: the preparation of the equilibrium solutions, why the spectrophotometric measurements were made at 260 nm and if the calibration curve was made, how the solubility and the exact amount of ketoprofen were calculated.

Lines 110-114 “For determination of the exact amount of ketoprofen used for in vitro dissolution study and for maintaining sink conditions of the poorly soluble drug, equilibrium solubility in different simulating gastric fluid (0.1 M HCl) containing three different concentrations (0.1, 0.5 and 1% w/v) of the anionic surfactant sodium lauryl sulphate as above mentioned.”. Please rephrase. It is not understood why sodium lauryl sulphate is added.

6.      Lines 115-121: Differential scanning calorimetry (DSC) and Fourier transfer infrared spectroscopy (FTIR). For DSC, please specify in which atmosphere the determinations were made and if the device was calibrated. For FTIR please specify the pressure of hydraulic press and the software used for data interpretation.

7.      In vitro dissolution: Lines 122-128: there are two repeated phrases: “the pH of the medium was raised to pH 6.8 using dibasic sodium phosphatein the same flask for additional 3 hours.” and “after 2 hours, the pH of the medium was raised to 6.8 using dibasic sodium phosphate for additional 3 hours”. Please correct.

8.      Please explain why used the USP 2 apparatus for dissolution in gelatin capsules. The gelatin capsules are the floating dose forms.

9.      Line 154: Writhing assay. After carboxymethyl cellulose should be written (CMC).

10.  Results. Solubility studies. The solubility curves for ketoprofen in aqueous solutions containing of 0.1 to 3% percentages of L-lysine, L-arginine and Tris are presented. The solubility of ketoprofen, physical mixtures (PM) and coprecipitated mixtures (Coppt) of ketoprofen in different simulating gastric fluid (0.1 M HCl) containing three different concentrations (0.1, 0.5 and 1% w/v) of the anionic surfactant sodium lauryl sulphate are not discussed.

11.  Results. Solubility studies. Lines 203-205: The sentence “Similarly, the solubility of ketoprofen from the prepared physical mixtures (PM) and coprecipitated mixtures (Coppt) of ketoprofen with Tris, L-lysine and L-arginine (Figure 3).” is not finished.

12.   Results. FTIR and DSC studies: Lines 220-222: “the characteristic peaks at their assignments wave numbers were simple addition to the FTIR spectra of the three basic excipients.” I understand this sentence. What is it referring to?

13.  Results. FTIR and DSC studies. You said that “DSC analysis indicated complete disappearance of ketoprofen melting from both physical and coprecipitated mixtures of ketoprofen with the three basic excipients.” In fig. 4C the melting peak of ketoprofen is present in the K:Lysine Coppt curve and in fig. 4D  the melting peak of ketoprofen is present in the K:Arginine PM curve. Please present in detail the interpretation of the DSC curves. Explain the appearance of new peaks or their total disappearance from the DSC curves.

14.  Results. Dissolution studies. Figure 5 should be presented in color and the range of pH used should be delimited. Coprecipitated (dispersed mixtures) are missing in the legend of figure 5.

15.  Results. Dissolution studies. Table 2.time required for dissolution of 50% of ketoprofen (T50%), relative dissolution rates at 60 min (RDR60 min) and relative dissolution rates at 300 min (RDR300 min)should be passed as the table legend.

16.  Results. Dissolution studies. The interpretation of the results must take into account the fact that after 120 min the pH of the solution is 6.8. Explain how the pH of the solutions and the pKa of the excipients influence the dissolution tests.

17.  Results. Writhing assay. Line 304: For K:L-arginine system Coppt is missing.

18.  The resolution of figure 10 should be improved.

Many typos have been found and should be corrected.

Author Response

The authors are deeply grateful to the reviewer for time and effort inveted to comment on the manuscript. We have happily addressed all comments and submitted the revised version of manuscript with the changes are highlighted as  track changes.

The paper “Exploration of the solubilizing, dissolution, analgesic effects and safety of common basic excipients on the NSAID drug ketoprofen” by  Heba A. Abou-Taleb, Mai E Shoman, Tarek Saad Makram, Jelan A. Abdel-Aleem and Hamdy Abdelkader reports  the complexation/salt formation of ketoprofen with the basic excipients (Tris, L-lysine and L-arginine). The mixtures were prepared using physical mixing and coprecipitation methods. The prepared mixtures were studied for solubility, docking, dissolution, DSC, FTIR, in vivo evaluation for efficacy (writhing test) and safety (ulcerogenic liability).

The following observations should be applied:

  1. Abstract. Line 23: “FIR” must be replaced with FTIR.

The abbreviation has been corrected accordingly.

  1. Material and methods: Physical mixtures: Lines 90-91: “Drug-excipient mixed thoroughly in a porcelain dish for 2-3 minutes using a spatula” is not enough to obtain a homogeneous mixture and especially a complex/salt formation. Please explain why used this method.

This method (physical mixing)  is normally used side by side with other techniques such as such as comelting  and coprecipitation for preparation of solid dispersions

  1. Material and methods: Coprecipitated dispersed mixtures: Lines 93-94: please specify the composition of the mixtures.

The composition of the dispersed mixtures has now been specified in details.

  1. Material and methods: Equilibrium solubility studies: Lines 100-101: “Excess amounts of ketoprofen were added to different solutions containing increasing concentrations (0, 0.1, 0.2, 0.4, 0.5, 1, 2 and 3% w/v)”. Increasing concentrations of what? Please specify.

The whole section has now been revised and corrected.   

  1. Material and methods: Equilibrium solubility studies: The whole paragraph needs to be rewritten. It must contain in detail: the preparation of the equilibrium solutions, why the spectrophotometric measurements were made at 260 nm and if the calibration curve was made, how the solubility and the exact amount of ketoprofen were calculated.

The whole paragraph has now been rewritten to show in more details.

Lines 110-114 “For determination of the exact amount of ketoprofen used for in vitro dissolution study and for maintaining sink conditions of the poorly soluble drug, equilibrium solubility in different simulating gastric fluid (0.1 M HCl) containing three different concentrations (0.1, 0.5 and 1% w/v) of the anionic surfactant sodium lauryl sulphate as above mentioned.”. Please rephrase. It is not understood why sodium lauryl sulphate is added.

These sentences have been moved to Section 3.3. The reasons of adding sodium lauryl sulphate were provided.

Sodium lauryl sulphate is an anionic surfactant that was selected because it mimics the anionic natural surfactants/bile salts in the gastric fluid. Wetting the dispersed particles is important in the dissolution process to avoid surface floating of drug particles and further mimic the in vivo performance, since it has been found that the surface tension of gastric fluid is considerably lower than that of water, suggesting the presence of surfactant in this region

  1. Lines 115-121: Differential scanning calorimetry (DSC) and Fourier transfer infrared spectroscopy (FTIR). For DSC, please specify in which atmosphere the determinations were made and if the device was calibrated. For FTIR please specify the pressure of hydraulic press and the software used for data interpretation.

The required information has now been provided.

  1. In vitro dissolution: Lines 122-128: there are two repeated phrases: “the pH of the medium was raised to pH 6.8 using dibasic sodium phosphatein the same flask for additional 3 hours.” and “after 2 hours, the pH of the medium was raised to 6.8 using dibasic sodium phosphate for additional 3 hours”. Please correct.

The whole section has been revised and corrected accordingly.

  1. Please explain why used the USP 2 apparatus for dissolution in gelatin capsules. The gelatin capsules are the floating dose forms.

USP apparatus 2 was more suitable than basket type especially we dealt with  powder mixtures and it require. For avoiding capsules floating, dissolution sinkers to  retain a solid dosage form at the bottom of the vessel.

  1. Line 154: Writhing assay. After carboxymethyl cellulose should be written (CMC).

The abbreviation CMC has been provided.

  1. Results. Solubility studies. The solubility curves for ketoprofen in aqueous solutions containing of 0.1 to 3% percentages of L-lysine, L-arginine and Tris are presented. The solubility of ketoprofen, physical mixtures (PM) and coprecipitated mixtures (Coppt) of ketoprofen in different simulating gastric fluid (0.1 M HCl) containing three different concentrations (0.1, 0.5 and 1% w/v) of the anionic surfactant sodium lauryl sulphate are not discussed.

The results were provided and discussed. Please refer to section 3.3.

Ketoprofen solubility recorded average values of 75, 127.5, 150 and 190 µg/ml for HCl solutions containing SLS 0.1, 0.5 and 1% w/v, respectively. Therefore, the acid dissolution medium with 1% of SLS was selected to ensure sink conditions.

  1. Results. Solubility studies. Lines 203-205: The sentence “Similarly, the solubility of ketoprofen from the prepared physical mixtures (PM) and coprecipitated mixtures (Coppt) of ketoprofen with Tris, L-lysine and L-arginine (Figure 3).” is not finished.

The sentence has now been completed.

  1. Results. FTIR and DSC studies: Lines 220-222: “the characteristic peaks at their assignments wave numbers were simple addition to the FTIR spectra of the three basic excipients.” I understand this sentence. What is it referring to?

The significance of these findings has been provided.

  1. Results. FTIR and DSC studies. You said that “DSC analysis indicated complete disappearance of ketoprofen melting from both physical and coprecipitated mixtures of ketoprofen with the three basic excipients.” In fig. 4C the melting peak of ketoprofen is present in the K:Lysine Coppt curve and in fig. 4D the melting peak of ketoprofen is present in the K:Arginine PM curve. Please present in detail the interpretation of the DSC curves. Explain the appearance of new peaks or their total disappearance from the DSC curves.

This section has now been rewritten and discussed accordingly.

Similarly, DSC analysis indicated complete disappearance of ketoprofen melting from both physical and coprecipitated mixtures of ketoprofen (K) with L-lysine and Tris. This indicated physicochemical electrostatic attractions. On the contrary, a weak melting transition was recorded with K:L-arginine PM but complete disapearnce of K melting with the coprecipitated mixtures

  1. Results. Dissolution studies. Figure 5 should be presented in color and the range of pH used should be delimited. Coprecipitated (dispersed mixtures) are missing in the legend of figure 5.

A colored version of Figure 5 has now been provided and the caption was modified to indicate that two dissolution media were used.

Coprecipitated mixtures are provided with abbreviation Coppt.

  1. Results. Dissolution studies. Table 2. “time required for dissolution of 50% of ketoprofen (T50%), relative dissolution rates at 60 min (RDR60 min) and relative dissolution rates at 300 min (RDR300 min)” should be passed as the table legend.

Table’s caption has now been modified accordingly.

  1. Results. Dissolution studies. The interpretation of the results must take into account the fact that after 120 min the pH of the solution is 6.8. Explain how the pH of the solutions and the pKa of the excipients influence the dissolution tests.

The results and discussion section3.3 have been modified accordingly.

   In this study, both acidic gastric conditions and physiological pH 6.8 simulating the intestinal pH were adopted. The initial two hours of dissolution were studied in acidic pH as the solubility of ketoprofen (the weak acidic drug, pKa=4.4) is very low (0.1mg/ml); because pH was well below pKa of ketoprofen and the drug was avialable in unionized species. On the contrary, the solubility of the drug at pH 6.8 (where the drug predominantly existed in ionized forms)  and evaluate the capacity of the three basic excipients to improve the dissolution rate under the gastric conditions. 

  1. Results. Writhing assay. Line 304: For K:L-arginine system Coppt is missing.

The whole section has been revised and discussed clearly.

  1. The resolution of figure 10 should be improved.

The resolution of Figure 10 has been improved to 300 DPI.

Many typos have been found and should be corrected.

The whole manuscript has now been revised and proofread extensively.

Reviewer 2 Report

The manuscript titled “Exploration of the solubilizing, dissolution, analgesic effects and safety of common basic excipients on the NSAID drug ketoprofen (Manuscript ID pharmaceutics-2209562) needs to be completed. Certainly, in vivo studies are an important part of this manuscript and add value to the research.

1)      I have a serious objection to the novelty of the research.

In the article (https://www.mdpi.com/2073-4352/12/2/275), published in 2022, was described the study of the dissolution rate of ketoprofen by preparing multicomponent crystals with tromethamine. Similarly, for ketoprofen and lysine, i.e. salt/cocrystal in (https://www.mdpi.com/1424-8247/14/6/555), and in (https://koreascience.kr/article/JAKO198203042496743.view). There are also known combinations and studies with ketoprofen and arginine

(https://pubchem.ncbi.nlm.nih.gov/compound/Ketoprofen-arginine).

Therefore, the paper should be developed in terms of previously published studies and clearly highlight what new results have been obtained. An in-depth analysis of literature data and an extended discussion of the results obtained by the authors together with a comparison with previous studies will be useful.

2)      I would like to know how is the thermal stability of ketoprofen at presence of TRIS and L-Lysine and L-Arginine (by DSC method).

3)      The numerical values ​​require magnification to improve the readability of the Figure 4.

4)      How was the atmosphere in the DSC experiments (nitrogen or other gas)?
Line 277, it is: „Pka” should be “pKa

5)      The reference list should be prepared by the ACS style guide.

The manuscript should be reconsidered after major revision, because it has potential. The paper needs refinement at the moment.

Author Response

The authors are sincerely grateful to the reviewer for the valuable comments. We have responded carefully to all comments and the manuscript is trackhanged to highlight the changes.

The manuscript titled “Exploration of the solubilizing, dissolution, analgesic effects and safety of common basic excipients on the NSAID drug ketoprofen (Manuscript ID pharmaceutics-2209562) needs to be completed. Certainly, in vivo studies are an important part of this manuscript and add value to the research.

1)      I have a serious objection to the novelty of the research.

In the article (https://www.mdpi.com/2073-4352/12/2/275), published in 2022, was described the study of the dissolution rate of ketoprofen by preparing multicomponent crystals with tromethamine. Similarly, for ketoprofen and lysine, i.e. salt/cocrystal in (https://www.mdpi.com/1424-8247/14/6/555), and in (https://koreascience.kr/article/JAKO198203042496743.view). There are also known combinations and studies with ketoprofen and arginine

(https://pubchem.ncbi.nlm.nih.gov/compound/Ketoprofen-arginine).

Therefore, the paper should be developed in terms of previously published studies and clearly highlight what new results have been obtained. An in-depth analysis of literature data and an extended discussion of the results obtained by the authors together with a comparison with previous studies will be useful.

The introduction and results sections have been modified accordingly. We have already reported these studies in our previous version but more in-depth coverage and highlights were added.

Our work is markedly different in that it studied the three excipients together ina comparative studies and investigated into analgesic activity using a pharmacodynamics model (writhing assay) and gastric safety index and histological documentation.

2)   I would like to know how is the thermal stability of ketoprofen at presence of TRIS and L-Lysine and L-Arginine (by DSC method).

DSC is a tool to investigate physicochemical properties and melting behaviours. No exothermic events were recorded throughout the study indicating the drug is stable during DSC study.

3)      The numerical values ​​require magnification to improve the readability of the Figure 4.

The resolution of Figure 4 has now been improved and an enhanced version was provided.

4)      How was the atmosphere in the DSC experiments (nitrogen or other gas)?

The data was provided about purging the system with nitrogen gas.

Line 277, it is: „Pka” should be “pKa”

It has now been corrected.

5)      The reference list should be prepared by the ACS style guide.

The references have been modified accordingly.

Round 2

Reviewer 1 Report

Comments and Suggestions for Authors

The paper “Exploration of the solubilizing, dissolution, analgesic effects and safety of common basic excipients on the NSAID drug ketoprofen” by  Heba A. Abou-Taleb, Mai E Shoman, Tarek Saad Makram, Jelan A. Abdel-Aleem and Hamdy Abdelkader reports  the complexation/salt formation of ketoprofen with the basic excipients (Tris, L-lysine and L-arginine). The mixtures were prepared using physical mixing and coprecipitation methods. The prepared mixtures were studied for solubility, docking, dissolution, DSC, FTIR, in vivo evaluation for efficacy (writhing test) and safety (ulcerogenic liability).

The following observations should be applied:

1.      Line 252:” no observable physicohcemcial” please correct it.

2.      Results. Solubility studies. Lines 271-274: The sentence “For determination of the exact amount of ketoprofen used for in vitro dissolution study under sink conditions, equilibrium solubility in different simulating gastric fluid (0.1 M HCl) containing three different concentrations (0.1, 0.5 and 1% 273 w/v) of sodium lauryl sulphate as a surfactant.” is not finished.

3.      Results. Solubility studies. Lines 284-290:” The initial two hours of dissolution were studied in acidic pH as the solubility of ketoprofen (the weak acidic drug) is very low (0.1mg/ml), pKa=4.4) is very low (0.1mg/ml); because pH was well below pKa of ketoprofen and the drug was avialable in unionized species. On the contrary, the solubility of the drug at pH 6.8 (where the drug predominantly existed in ionized forms) and evaluate the capacity of the three basic excipients to improve the dissolution rate under the gastric conditions.”. I don’t understand this sentence. Please rephrase.

Author Response

The following observations should be applied:

  1. Line 252:” no observable physicohcemcial” please correct it.

It has been corrected.

  1. Solubility studies. Lines 271-274: The sentence “For determination of the exact amount of ketoprofen used for in vitro dissolution study under sink conditions, equilibrium solubility in different simulating gastric fluid (0.1 M HCl) containing three different concentrations (0.1, 0.5 and 1% 273 w/v) of sodium lauryl sulphate as a surfactant.” is not finished.

The whole paragraph has been revised

  1. Solubility studies. Lines 284-290:” The initial two hours of dissolution were studied in acidic pH as the solubility of ketoprofen (the weak acidic drug) is very low (0.1mg/ml), pKa=4.4) is very low (0.1mg/ml); because pH was well below pKa of ketoprofen and the drug was avialable in unionized species. On the contrary, the solubility of the drug at pH 6.8 (where the drug predominantly existed in ionized forms) and evaluate the capacity of the three basic excipients to improve the dissolution rate under the gastric conditions.”. I don’t understand this sentence. Please rephrase.

The whole paragraph and the manuscript have been revised for English Language by a professional proof-reader. The certificate is kindly attached.

Reviewer 2 Report

Thank you for completing the manuscript. I recommend the paper for publication.

Author Response

Thank you for completing the manuscript. I recommend the paper for publication.

Thank you for that. The whole manuscript has been revised for English Language by a professional proof-reader. The certifcate is kindly attached.
